# Imit-Diff: Semantics Guided Diffusion Transformer with Dual Resolution Fusion for Imitation Learning

## Abstract

Diffusion-based methods have become one of the most important paradigms in the field of imitation learning. However, even in state-of-the-art diffusion-based policies, there has been insufficient focus on semantics and fine-grained feature extraction, resulting in weaker generalization and a reliance on controlled environments. To address this issue, we propose **Imit-Diff**, which consists of three key components: **1) Dual Resolution Fusion** for extracting fine-grained features with a manageable number of tokens by integrating high-resolution features into low-resolution visual embedding through an attention mechanism; **2) Semantics Injection** to explicitly incorporate semantic information by using prior masks obtained from open vocabulary models, achieving a world-level understanding of imitation learning tasks; and **3) Consistency Policy on Diffusion Transformer** to reduce the inference time of diffusion models by training a student model to implement few-step denoising on the Probability Flow ODE trajectory. Experimental results show that our method significantly outperforms state-of-the-art methods, especially in cluttered scenes, and is highly robust to task interruptions. The code will be publicly available.

## 1 Introduction

Imitation learning (Zhao et al., 2023; Bonardi et al., 2020; Cheng et al., 2024; Dasari & Gupta, 2021; Englert & Toussaint, 2018; He et al., 2024; Luo et al., 2023; Fu et al., 2024; Team et al., 2024; Wu et al., 2024b) provides an efficient framework for robots to acquire human skills by leveraging expert demonstrations. Existing methods, which typically follow a supervised learning paradigm, use either explicit (Torabi et al., 2018) or implicit policies to map the robot's observations to the action space or its latent representation space. These methods often rely on approaches such as mixtures of Gaussians (Zhao et al., 2023) or categorical representations (Lee et al., 2024) of discretized action to approximate the action distribution. However, such techniques generally generate action sequences through a single forward pass, limiting their expressiveness in high-dimensional spaces and constraining their ability to accurately capture the complexity of multimodal action distributions (Chi et al., 2023). Moreover, the reliance on one-shot generation makes these models vulnerable to noise and outliers, undermining their robustness in real-world applications.

Diffusion models (Chen, 2023; Chen et al., 2023; Chi et al., 2023; Fan et al., 2024; Huang et al., 2023b; Mishra et al., 2023; Ze et al., 2024), which employ a conditional denoising diffusion process for visuomotor policy learning, have demonstrated remarkable effectiveness in tackling complex, robotic tasks. The Diffusion Transformer architecture, DiT (Peebles & Xie, 2023), leverages the attention mechanism to capture global context. It is highly effective at modeling long-range dependencies, which makes it particularly well-suited for handling both vision and action sequences in robotic applications. As a result, this architecture has emerged as a dominant paradigm in diffusion models. However, when using conditional embeddings to guide the denoising of action sequences, existing diffusion-based methods lack effective extraction of fine-grained features as shown in Fig. 1. On the other hand, although previous works (Huang et al., 2023a;c; Li et al., 2024a; Yu et al., 2023) have attempted to introduce high-level semantic information to supervise agents in completing tasks, they have not explored methods for incorporating fine-grained semantic information to

Figure 1: Comparison of current imitation learning paradigms. (a) **ACT Like Policy** refers to the method of directly mapping robot observation to action space through a feedforward with the challenge of weak representation for complex distributed actions. (b) **Diffusion Like Policy** extracts observation as a conditional vector to supervise the iterative denoising of action sequence with insufficient focus on feature representation. (c) Our method **Imit-Diff** introduces dual res fusion for fine-grained capture and prior mask for semantic information to raise world-level understanding.

capture subtle variations. This poses challenges for embodied intelligence in understanding scenes and tasks.

To tackle these challenging problems, we introduce **Imit-Diff**, an imitation learning policy network that explicitly incorporates prior-based semantics and enhances the representation of observation features to improve the robot's fine-grained perception and scene understanding. Specifically, the model extracts detailed information from the scene through high and low-resolution fusion. Additionally, we use open vocabulary models to introduce prior masks, which explicitly capture and align semantic information. Furthermore, we implement a Consistency Policy for the Diffusion Transformer, effectively increasing the robot's action execution frequency.

In conclusion, our contributions are three-fold:

1) Dual Resolution Fusion to improve fine-grained feature representation.

2) Semantics Injection to introduce semantics information with prior masks obtained through open vocabulary models.

3) Implementation of Consistency Policy for Diffusion Transformer to reduce inference time for DiT-based models.

The experiments demonstrate that our method works effectively and the code will be open-source soon.

## 2 RELATED WORK

### 2.1 DIFFUSION POLICY IN IMITATION LEARNING

Diffusion models, a category of generative models that progressively sample data from random noise, have gained significant traction and impressive expressiveness in robotic applications. In the context of robotics learning, diffusion models are utilized as effective policy networks for imitation learning. For instance, Diffusion Policy (Chi et al., 2023) aggregates observations into a conditional embedding to guide the denoising process of action sequences. However, compressing diverse observation information into a single embedding can lead to information loss. Subsequent work such as UIM (Kaewpoonsuk & Subsomboon, 2024), extended the conditional information from a single embedding to a token sequence, but it didn't adequately address the integration of robot proprioceptive states with environmental observations. Recent advances, like Aloha Unleashed (Zhao et al.), expanded the Hybrid Transformer architecture from the ACT algorithm into Diffusion Policy. However real-world robotic systems often require more sophisticated data mining and integration methods to handle complex scenarios. In our work, we leverage a dual-resolution encoder to fuse high and low-resolution features. We also utilize prior masks to guide the attention mechanism to focus

on critical areas, thereby enhancing the scene understanding and fine-grained extraction in imitation learning.

## 2.2 Acceleration Strategies for Diffusion Models in Robotics

As mentioned in Sec. 1, diffusion models come with certain drawbacks, including long inference times due to their iterative sampling process. Given the real-time requirements of applications in robotics, such as robot control, accelerating diffusion models is a critical issue for improving performance. One line of work, such as DDIM (Han, 2024) and EDM (Hasan et al., 2023), can be interpreted as integrating deterministic ODEs (Zheng et al., 2023), addressing the long inference times by reducing the number of denoising steps for prediction. However, while this variable-step approach reduces the number of denoising steps, it can also degrade sample quality. Another line of research aims to accelerate diffusion models through parallel sampling, using methods like Picard iteration (Han et al., 2024; Andrade et al., 2023; Wang et al., 2024b), which attempt to converge batches of points along the diffusion ODE trajectory in parallel. Due to the significant increase in memory demand caused by this parallelization, such methods are impractical in computationally constrained robotic settings. Distillation-based techniques (Wu et al., 2024a; Guo et al., 2023; Wang et al., 2023; Phuong & Lampert, 2019; Hao et al., 2024; Gou et al., 2021) train new student models from pre-trained teacher models, allowing the student to take larger steps along the ODE trajectory that the teacher has already mapped. The Consistency Policy (Prasad et al., 2024) introduced by Aaditya et al. allows student models to map inputs at arbitrary step sizes and intervals to the same starting point on the given ODE trajectory, demonstrating superiority in robot control tasks within the U-Net (Ronneberger et al., 2015) architecture. In our work, we implement the CTMs framework on top of the Diffusion Transformer, validating its orthogonality to the policy learning framework. This resulted in an order-of-magnitude improvement in inference speed, which allows us to use temporal ensemble and action dropout to enhance real-time performance and smoothness.

## 2.3 Open Vocabulary Vision Foundation Models

Open vocabulary vision foundation models (Liu et al., 2023; Ren et al., 2024a;b; Wasim et al., 2024; Yuen et al., 2024) enable the understanding of images through vision-language learning, allowing natural language descriptions to guide visual comprehension. These models generalize well across various downstream tasks and can be used in robotics as tools for defining complex goals, semantic anchors for multimodal representation, and intermediate substrates for planning and reasoning. Although end-to-end methods are popular in offline tasks, learning directly from language-annotated data presents challenges, particularly in mapping language, visual observations, and robotic sensor data into a shared space. In this work, we use open vocabulary vision foundation models to translate language into vision obervation for key object identification in robotic manipulation. Grounding DINO (Liu et al., 2023) is employed for detection, combined with a MixFormerV2-based (Cui et al., 2022; 2024) multi-object tracker for real-time performance and occlusion handling. Mobile SAM (Zhang et al., 2023) is used to segment target objects, providing RGB-MASKs (Wang et al., 2024a) as observations for the policy network.

## 3 Method

The proposed method Imit-Diff mainly consists of four parts: Dual Resolution Fusion (see Sec. 3.1) to enhance representation capacity of visual tokens, Semantics Injection (see Sec. 3.2) to involve prior knowledge to aid environment perception, Consistency Policy within DiT to accelerate inference and Temporal Optimization (see Sec. 3.3).

### 3.1 Dual Resolution Fusion

In the methodology of imitation learning, models are trained to predict actions given sequential observations from the environment. Since the time intervals between the observations are relatively small, the ability to perceive fine-grained details in high-resolution observations is of vital importance. However, in previous methods, the environment observations are either transformed to low-dimension feature vectors via a CNN network (thus losing fine-grained details) (Zhao et al., 2023), or directly down-sampled to lower resolution (224x224 in Chi et al. (2023)).

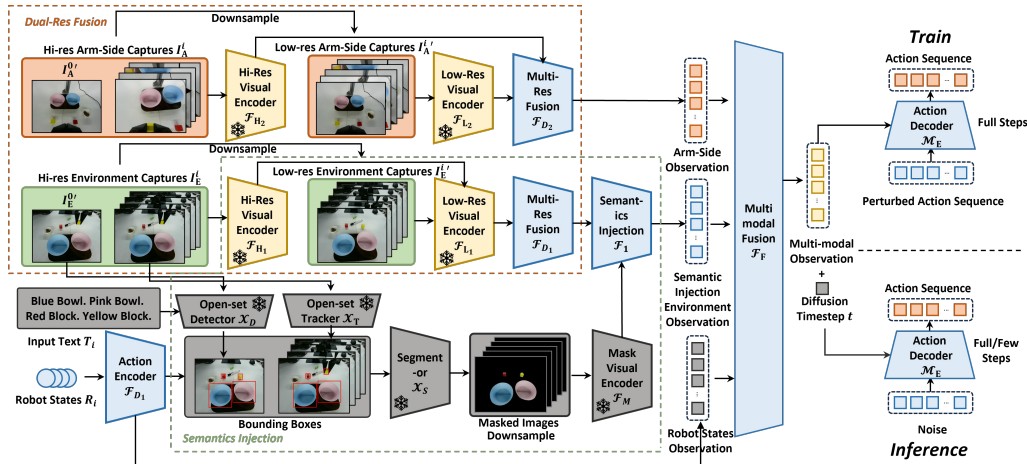

Figure 2: **Overview of Imit-Diff** that consists of *1)* **Dual-Res Fusion**: High-resolution images are downsampled to obtain low-resolution images, which are passed through a vision encoder for multi-resolution fusion. This process encodes visual embeddings with fine-grained information. *2)* **Semantics Injection**: High-resolution images are processed by open vocabulary models to generate masks. We use the same pretrained encoder as the low-res visual encoder to extract mask features, which are then injected into the multi-resolution fusion tokens to explicitly introduce semantic priors. *3)* **Consistency Policy for Diffusion Transformer**: Visual tokens are fused with robot state tokens in a multi-modal manner, guiding the denoising process of the action sequence.

One possible solution to address this problem is to use high-resolution environment captures to train the policy network, but the unacceptable increase in memory footprint and the difficulty of directly modeling high-dimensional image spaces make this solution impractical. Motivated by Li et al. (2024b), we propose **Dual Resolution Fusion** (illustrated in the orange boxes in Fig. 2), which incorporates both hi-res and low-res features when representing environmental observations with the same amount of tokens. In this way, the model is expected to understand the environment in multiple granularities, providing adaptive information when decoding action sequences.

Specifically, given high-resolution observations from environment cameras and arm-side cameras (the $i$-th frame denoted by $I_E^i$ and $I_A^i$ respectively), down-sampling is first applied to generate low-resolution observations $I_E^{i}{}'$ and $I_A^{i}{}'$. Then, the high-resolution and low-resolution observations are processed by pre-trained hi-res visual encoder $\mathcal{F}_H$ (implemented by ConvNext by Liu et al. (2022) followed by feature pyramid networks) and pre-trained low-res visual encoder $\mathcal{F}_L$ (implemented by ViT-S version of DINOv2 by Oquab et al. (2023)). After being projected to the same dimension, the features are fused by the self-attention layer $\mathcal{F}_D$ whereas low-res features are regarded as queries and high-res features are regarded as keys and values. Note that, the parameters of $\mathcal{F}_H$ and $\mathcal{F}_L$ are frozen during training while $\mathcal{F}_D$ is optimized during training.

This design allows the extraction of high-resolution details without drastically increasing the number of tokens for diffusion policy inference, thereby enhancing scene understanding with an acceptable length of conditional sequence.

## 3.2 SEMANTICS INJECTION

Although massive progress has been achieved by previous imitation learning methods (e.g. Fu et al. (2024), Zhao et al.), the current models are only able to perform specific tasks under a carefully controlled environment. This could be ascribed to the limited amount of demonstrations or the over-fitting in the latent space as the demonstrations are collected in an almost unchanged environment. To overcome this limitation, world-level knowledge embedded in the pre-trained multi-modal models could be used to prevent unnecessary focus on task-irrelevant details. The grounding of knowledge into the provided environment could be achieved by performing open-set detection and segmentation, which we call **Semantics Injection** (illustrated in the green boxes of Fig. 2).

To perform **Semantics Injection**, the task-relevant phrases (e.g. red bowls) and the first frame of the downsampled environment capture $I_E^{0\prime}$ are fed into an open-set detector $\mathcal{X}_D$ (implemented by Grounding DINO Liu et al. (2023)) to obtain relevant bounding boxes. To assure temporal consistency, the subsequent frames are processed via an end-to-end tracking model $\mathcal{X}_T$ (implemented by MixFormerv2 by Cui et al. (2024)) given the latest predicted bounding boxes and captured frames. Subsequently, an open-set segmentation model $\mathcal{X}_S$ (implemented by MobileSAM by Zhang et al. (2023)) is used to provide semantic masks and later semantic masked images.

Then, the injection of semantics is performed by fusing the feature extracted from mask visual encoder $\mathcal{F}_M$ (implemented by ViT-S version of DINOv2 by Oquab et al. (2023)) and multi-resolution features extracted by $\mathcal{F}_{D_1}$ with $\mathcal{F}_I$, a transformer decoder with masked image features used as queries. The semantic injected environment observation (output of $\mathcal{F}_I$) is later concatenated with arm-side observations and robot state observations (generated by action encoder $\mathcal{F}_A$, a multi-layer perceptron), which is then fed to multi-modal fusion module $\mathcal{F}_F$ (a transformer encoder) to perform cross-modal fusion.

### 3.3 CONSISTENCY POLICY FOR DIFFUSION TRANSFORMER

Prasad et al. (2024) proposed U-Net-based Consistency Policy, allowing the prediction of action sequences with few-step or single-step diffusion. In the consistency policy method, the teacher model, denoted as $s_\phi$, is trained under the EDM framework whereas the student model is distilled from the teacher model. The EDM framework takes the current position $\mathbf{x}_t$, time $t$, and condition $o$ as input to estimate the derivative of the Probability Flow ODE (PFODE) trajectory:

$$\mathrm{d}\mathbf{x}_t/\mathrm{d}t = -\left(\mathbf{x}_t - s_\phi\left(\mathbf{x}_t, t; o\right)\right)/t, \tag{1}$$

where $\mathbf{x}_t$ denotes the general form of the PFODE:

$$\mathrm{d}\mathbf{x}_t = \left[\boldsymbol{\mu}\left(\mathbf{x}_t, t\right) - \frac{1}{2}\sigma(t)^2 \nabla \log p_t\left(\mathbf{x}_t | o\right)\right] \mathrm{d}t. \tag{2}$$

The optimized Denoising Score Matching (DSM) loss is used to train the EDM model:

$$\mathcal{L}_{\mathrm{DSM}}(\boldsymbol{\theta}) = \mathbf{E}_{t, \mathbf{x}_0, \mathbf{x}_t | \mathbf{x}_0} \left[d\left(\mathbf{x}_0, s_\phi(\mathbf{x}_t, t; o)\right)\right]. \tag{3}$$

The DSM objective samples a point along the PFODE, $(\mathbf{x}_t, t)$, and trains the EDM model to predict the ground truth initial position $\mathbf{x}_0$. Unlike the Consistency Policy, we use MSE Loss instead of the pseudo-Huber Loss as $d(\cdot, \cdot)$, giving higher weight to smaller fine-grained action errors.

$$d(x, y) = \|x - y\|_2^2. \tag{4}$$

The student model $g_\theta(\mathbf{x}_t, t, s; o)$ samples two positions $\mathbf{x_t}$ and $\mathbf{x_u}$ on the same PFODE, and denoises both positions back to the same time step $s$. After calculating $g_\theta(\mathbf{x}_t, t, s; o)$ and $g_\theta(\mathbf{x}_u, u, s; o)$, we use $g_\theta(\mathbf{x}_s^{(t)}, s, 0; o)$ and $g_\theta(\mathbf{x}_s^{(u)}, s, 0; o)$ to bring these two samples, referred to as $\mathbf{x}_s^{(t)}$ and $\mathbf{x}_s^{(u)}$, back to time 0. The loss is always measured in the fully denoised action space:

$$\mathcal{L}_{CTM} = d(g_\theta(\mathbf{x}_s^{(t)}, s, 0; o), g_\theta(\mathbf{x}_s^{(u)}, s, 0; o)). \tag{5}$$

The final training objective combines DSM Loss and CTM Loss:

$$\mathcal{L}_{CP} = \alpha \mathcal{L}_{CTM} + \beta \mathcal{L}_{DSM}. \tag{6}$$

In practice, we implement the Consistency Policy by Prasad et al. (2024) on the backbone of Diffusion Transformer, originally proposed by Peebles & Xie (2023). Specifically, as illustrated in Fig. 2, the time step is concatenated with the output of the multi-modal fusion module, which is later used as the condition embedding for consistency policy denoising. With time step concatenated, the

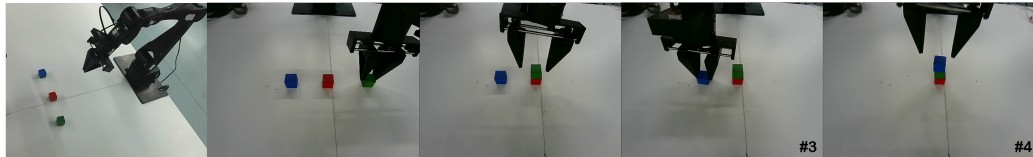

**Block Placement:** Subtask **#1** & **#2** Pre-Grasp | Subtask **#3** Grasp Block | Subtask **#4** Place Block

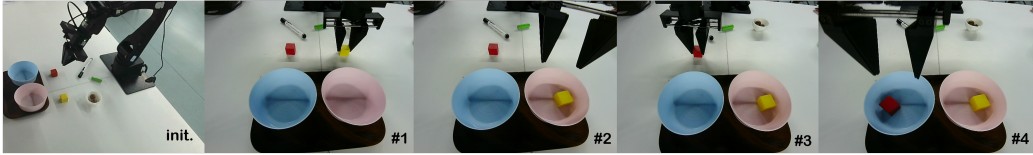

**Stack Blocks:** Subtask **#1** Grasp Block1 | Subtask **#2** Stack Block1| Subtask **#3** Grasp Block2 | Subtask **#4** Stack Block2

**Object Sortation:** Subtask **#1** Grasp Block1 | Subtask **#2** Place Block1| Subtask **#3** Grasp Block2 | Subtask **#4** Place Block2

Figure 3: A brief description of each real-world task in stages. For detailed task setup, see 4.1.

attention mechanism can be used to denoise action sequences, replacing the FiLM module used in U-Net. Furthermore, we hope to employ the Temporal Ensemble introduced by ACT to improve the smoothness and dynamic response of diffusion models by accelerating inference.

# 4 EXPERIMENTS

We conduct experiments to evaluate the performance of Imit-Diff in fine-grained manipulation tasks in complex scenarios. We design three real-world tasks to verify the advanced capabilities of Imit-Diff. ablation study is also conducted to demonstrate the effectiveness of each component of the proposed method.

## 4.1 TASK SETUP AND METRICS

**Tasks:** We evaluate Imit-Diff in the real world on three tasks: Block Placement, Object Sorting, and Stack Blocks (See Fig. 3). The settings for testing the model's anti-interference and generalization capabilities are shown in Fig. 4 in Appendix A.1.

**1) Block Placement:** In this task, the robot is expected to place a block in a bowl, while the cup is used as an obstacle that the robot would have to move away first. Other irrelevant clutters are also randomly placed in the scenes for deliberate interference. The relevant objects are termed as "blue cup", "green block" and "pink bowl". This task intends to evaluate the essential ability for the execution of the robot.

**2) Object Sortation:** With two blocks randomly placed on a plate with complex textures, the robot is expected to pick up the blocks further to the left and the right and place them in the left and the right bowls respectively. Irrelevant clutters are also randomly placed in the scenes for deliberate interference. The relevant objects are termed "yellow block", "red block", "blue bowl", and "pink bowl". This task intends to evaluate the robot's robustness against cluttered scenes.

**3) Stack Blocks:** With three blocks placed on the desk, the robot is expected to stack three blocks sequentially. Irrelevant clutters are also randomly placed in the scenes for deliberate interference. The relevant objects are termed as "green block", "blue block", and "red block". This task intends to evaluate the manipulation precision.

For the aforementioned tasks, we use a 6-DoF AIRBOT Play robot arm for collecting expert demonstrations with teleoperation. For each task, we collect only 50 demonstrations. During the demon-

strations, two USB cameras are used to capture RGB observations from different perspectives: two cameras mounted on the table and at the end of the robotic arm, respectively. We use $224 \times 224$ images as the low-resolution input and $448 \times 448$ images for high-resolution input. For fairness in comparison, we use the original image size of $480 \times 640$ as input for Diffusion Policy and ACT to ensure the preservation of raw information. The low-dimensional observations consist of observed joint positions, including the six joint positions of the robot arm and the gripper's position. We perform inference using a laptop with a single 4060 GPU and 8GB of VRAM. Notably, we adopt DDIM as the diffusion strategy for Diffusion Policy, using 16 steps for policy inference, which is consistent with the original implementation.

In terms of the metrics, we assess the robot's performance by the average success rate. We run 20 evaluations for each task and divide the task into several sub-tasks to assess the algorithm's predictions. For the target objects, we change their appearance without altering their geometric properties to evaluate the model's generalization of appearance.

### 4.2 BASELINES

We benchmark Imit-Diff against state-of-the-art imitation learning methods that have shown significant success in policy learning for complex robotic tasks. Specifically, we use ACT and Diffusion Policy as baseline models. Both ACT and Diffusion Policy employ the ResNet-18 vision backbone, as detailed in their original implementations. Similar to Imit-Diff, the baselines use a transformer architecture, and hyper-parameters such as prediction horizon and image resolution are are tuned similarly for a fair comparison. By comparing with baselines that have already demonstrated strong performance on complex tasks, our goal is to demonstrate that the introduction of prior mask-guided dual-vision fusion can improve generalization to clutter and fine-grained scene understanding within limited data. See Tab. 6 and Tab. 7 for training details.

### 4.3 RESULTS

We report the success rates of Imit-Diff and the baselines in Tab. 1. Imit-Diff achieved a success rate of 0.9 for Block Placement, 0.9 for Object Sorting, and 0.95 for Stack Blocks, outperforming both ACT and Diffusion Policy. The excellent performance on the fine operations (e.g., picking up and stacking blocks) demonstrates the benefits of the fine-grained feature extraction enabled by multi-resolution fusion.

In Tab. 2, we report the success rates of various methods in environments with clutter interference. The outstanding experimental results demonstrate that the introduction of the prior mask effectively improves generalization against interference.

Tab. 3 presents the robustness of different models against appearance changes. We replace the target objects with colors unseen during training, and Imit-Diff, unlike ACT and Diffusion Policy, is able to clearly identify the objects that should be attended to.

Notably, Tab. 3 also demonstrates the re-completion ability of each model. After the robot completes the tasks, we manually restore the scene to an intermediate sub-task state. Imit-Diff enables the robot to reassess the current scene and successfully complete the task again, regardless of object appearance. This demonstrates that the high-quality feature tokens constructed by Imit-Diff enhance scene understanding.

### 4.4 ABLATION STUDY

We aim to validate our design choices through several ablation studies and gain a better understanding of how different hyper-parameters influence Imit-Diff. We choose the most challenging real-world task for fine-grained feature extraction, Stack Blocks, as the benchmark for the ablation study.

Tab. 4 a) presents the results of ablations on visual backbones. We found that ViT-S DINOV2 significantly outperforms a simple ViT-S pretrained on ImageNet. This suggests that the pretrained weights have a crucial impact on the scene understanding capabilities of Imit-Diff. The superior performance of ViT-S DINOV2 can be attributed to its self-supervised pretraining, which enables it to learn rich, generalizable feature representations.

Tab. 4 b) shows the success rates for the Stack Blocks task under different loss designs. We find that the model performs better with MSE Loss, which is more commonly used in diffusion models, compared to Huber Loss used in Consistency Policy. This may be due to Huber Loss's higher tolerance for noise in tasks requiring fine manipulation, which can cause small action variations to be disregarded, while MSE Loss is more effective at capturing and reflecting these subtle movements.

In Tab. 4 c) , we present the results of the ablation study on camera views. We find that adding the arm-side view improves our model's performance in fine manipulation tasks, such as block stacking. It demonstrates that our network is scalable and can further enhance its performance by incorporating additional observational information due to multi-modal fusion.

Tab. 4 d) presents the results of our ablation study on semantics injection. The experimental setup is similar to that in Sec. 4.1. The results show that the model performs better, especially under unseen clutter interference, with the introduction of the prior mask. This validates the effectiveness of the component we proposed in Sec. 3.1.

In Tab. 4 e) , we present the results of the ablation study on dual-resolution fusion. As we progressively reduce the number of FPN layers described in Sec. 3.1, the model's performance also decreases, demonstrating the soundness of the component design in Sec. 3.1. Additionally, we identify FPN=3 as a sweet spot, balancing model performance and training cost.

### 4.5 CONSISTENCY POLICY WITH ACTION DROPOUT

In previous experiments, we have demonstrated the strong performance and generalization capabilities of our method. However, similar to other diffusion-based imitation learning algorithms, Imit-Diff suffers from longer inference times due to the EDM denoising framework. In Sec. 3.3, we introduce the implementation of the Consistency Policy within the DiT architecture. Tab. 5 reports the inference times of our model. The implementation of the Consistency Policy in the DiT significantly improves inference speed, making it possible to enhance dynamic responsiveness through Temporal Ensemble and Action Dropout, a method we designed to increase execution frequency by selectively dropping certain actions.

Table 1: Success rate (%) of 3 real-world tasks within 20 evaluation trials each, comparing our method with the two baselines. The model is trained with human demonstrations and fixed seed. Overall, Imit-Diff significantly outperforms previous methods.

| Method | Block Placement | | | Object Sortation | | | | Stack Blocks | | | |
| | Pre-Grasp | Grasp Block | Place Block | Grasp Block1 | Place Block1 | Grasp Block2 | Place Block2 | Grasp Block1 | Stack Block1 | Grasp Block2 | Stack Block2 |
| --- | --- | --- | --- | --- | --- | --- | --- | --- | --- | --- | --- |
| ACT | 95 | 90 | 100 | 90 | 95 | 100 | 100 | 95 | 95 | 100 | 90 |
| DP-T | 90 | 85 | 95 | 90 | 95 | 85 | 90 | 85 | 95 | 90 | 95 |
| Imit-Diff | 95 | 95 | 100 | 95 | 100 | 100 | 95 | 95 | 100 | 100 | 100 |

Table 2: Success rate (%) of 3 real-world tasks within 20 evaluation trials each **in cluttered scenes**. We compare the models' performance with clutters seen / unseen during training placed at random positions.

| Method | Block Placement | | Object Sortation | | Stack Blocks | |
| | Clutter Seen | Clutter Unseen | Clutter Seen | Clutter Unseen | Clutter Seen | Clutter Unseen |
| --- | --- | --- | --- | --- | --- | --- |
| ACT | 85 | 70 | 80 | 75 | 95 | 85 |
| DP-T | 80 | 65 | 85 | 75 | 90 | 80 |
| Imit-Diff | 95 | 90 | 95 | 90 | 95 | 95 |

Table 3: Success rate (%) of 3 real-world tasks within 20 evaluation trials each **with seen / unseen object appearance** and **with / without process interference**. Process interference refers to manually impeding after the task is done so that the model would have to restart from the intermediate stage.

| | Block Placement | | |
|---|---|---|---|
| Method | Appearance Seen | Appearance Seen + Process Interference | Appearance Unseen + Process Interference |
| ACT | 85 | 50 | 45 |
| DP-T | 75 | 60 | 40 |
| Imit-Diff | 90 | 90 | 80 |
| | Object Sortation | | |
| Method | Appearance Seen | Appearance Seen + Process Interference | Appearance Unseen + Process Interference |
| ACT | 85 | 75 | 70 |
| DP-T | 65 | 60 | 55 |
| Imit-Diff | 90 | 90 | 80 |
| | Stack Blocks | | |
| Method | Appearance Seen | Appearance Seen + Process Interference | Appearance Unseen + Process Interference |
| ACT | 80 | 85 | 80 |
| DP-T | 70 | 85 | 70 |
| Imit-Diff | 95 | 85 | 85 |

Table 4: Success rate (%) of the Stack Blocks task in ablation studies within 20 evaluation trials each.

| a). Visual Backbones | | b). Loss Designs | |
|---|---|---|---|
| ViT-S | ViT-S DINOV2 | Huber Loss | MSE Loss |
| 30 | 95 | 10 | 95 |

| c). Camera Views | |
|---|---|
| Env. View | Env. + Arm-side View |
| 90 | 95 |

| d). Semantics Injection | | |
|---|---|---|
| | With Semantics | Without Semantics |
| No Clutters | 95 | 95 |
| With Seen Clutters | 95 | 95 |
| With Unseen Clutters | 95 | 85 |

| e). Dual Resolution Fusion | | | | |
|---|---|---|---|---|
| FPN-0 | FPN-1 | FPN-2 | FPN-3 | FPN-4 |
| 20 | 30 | 60 | 85 | 95 |

Table 5: Inference Time of EDM and CTM Frameworks for Imit-Diff

| EDM | CTM (Single-step) | CTM (Few-step) |
|---|---|---|
| 1.5s | 0.06s | 0.12s |

## 5 LIMITATIONS AND CONCLUSIONS

**Conclusions:** We propose an imitation learning strategy for enhancing fine-grained feature representation and scene understanding, including improving fine-grained manipulation through dual-resolution fusion and introducing semantics through prior masks. The synergy between these two parts enables the model to obtain generalization against interference and learn fine operations, such as completing tasks in cluttered scenes and re-complete tasks from a certain stage.

**Limitations and Future Work:** Although our work outperforms on challenging tasks and shows excellent generalization, there are still practical issues of algorithmic capabilities and robotics engineering. Specifically, our approach based on the EDM framework suffers from long inference time. Although we have increased the inference speed by an order of magnitude in DiT to improve dynamic response, there is still a gap in running speed compared to lightweight algorithms such as ACT. In the future, we will explore the multi-modal fusion of robot observations including touch or 3D information. Overall, we hope that this representation-enhanced imitation learning algorithm can take an important step forward in robot perception and open-source resources.

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

# A APPENDIX

## A.1 EXPERIMENT DETAILS

Fig. 4 shows the experimental settings for model anti-interference and generalization capabilities.

## A.2 TRAINING DETAILS

All models are trained using the same collected data on a platform with $8 \times$ A100 GPUs. The training parameter settings of the baseline models are shown in Tab. 6, Tab. 7 and Tab. 8.

Table 6: ACT Training

| Hyperparameter | Value |
|---|---|
| input image shape | $3 \times 480 \times 640$ |
| learning rate | 2e-4 |
| batch size | 16 |
| steps | 10000 |
| feedforward dimension | 3200 |
| hidden dimension | 512 |
| chunk size | 100 |
| beta | 10 |
| dropout | 0.1 |

Table 7: Diffusion Policy Training

| Hyperparameter | Value |
| --- | --- |
| input image shape | $3 \times 448 \times 448$ |
| learning rate | 2e-4 |
| batch size | 64 |
| steps | 20000 |
| chunk size | 20 |
| scheduler | DDIM |
| train and test diffusion steps | 100,16 |
| ema power | 0.75 |
| backbone | pretrained ResNet18 |
| noise predictor | Transformer |

Table 8: Imit-Diff Training

| Hyperparameter | Value |
| --- | --- |
| high resolution image shape | $3 \times 448 \times 448$ |
| low resolution image shape | $3 \times 224 \times 224$ |
| learning rate | 1e-4 |
| batch size | 64 |
| steps | 20000 |
| chunk size | 20 |
| scheduler | EDM |
| train and test diffusion steps | 80,80 (EDM) — 3 (CTM) |
| ema power | 0.75 |
| backbone | pretrained ViT DINOV2 (LR) & pretrained ConvNext-Base (HR) |
| noise predictor | Transformer |

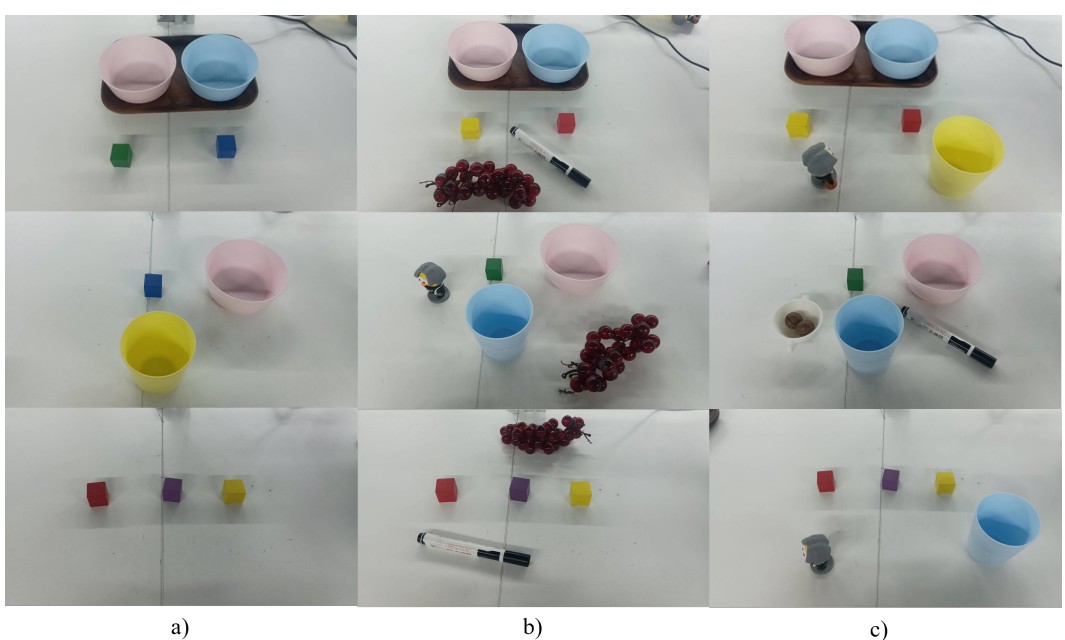

a)                                    b)                                    c)

Figure 4: Experimental settings for model anti-interference and generalization capabilities. To verify the model's ability to adapt to scenes with unseen manipulating objects and interfering objects, we set up multiple groups of experiments for each task: a) randomly changing the color of the manipulated objects in the task; b) randomly placing objects that exist in the training data; c) randomly placing objects that do not exist in the training data.

