# OpenReview forum: "Imit-Diff: Semantics Guided Diffusion Transformer with Dual Resolution Fusion for Imitation Learning"
_ICLR.cc/2025/Conference — ICLR 2025 Conference Withdrawn Submission_

### Official Review · Reviewer_749E · 2024-10-27

**Soundness:** 2
**Presentation:** 3
**Contribution:** 2
**Rating:** 3
**Confidence:** 4

**Summary:**

This paper presents Imit-Diff, an imitation learning model that enhances fine-grained perception and semantic understanding in robotics. To generalize in complex scenes, Imit-Diff introduces a *Dual Resolution Fusion* mechanism to integrate high- and low-resolution visual information, a *Semantics Injection* method to incorporate prior knowledge through masks from open vocabulary models, and a *Consistency Policy* that reduces inference time with an accelerated denoising process. Experimental results demonstrate that Imit-Diff achieves state-of-the-art performance on real-world tasks and outperforms current baselines like ACT and Diffusion Policy.

**Strengths:**

1. The proposed method integrates prior-based semantics to enhance generalization in imitation learning.
2. Using multi-scale features to improve the performance.

**Weaknesses:**

1. While semantic injection is the key contribution aimed at improving generalization, the results in Tab 4e do not demonstrate substantial benefits from this method.
2. Sec 3.3 allocates significant space to describing the consistency policy, yet it primarily introduces an existing policy without addressing specific challenges in applying it to the proposed algorithm. It would be helpful to clarify any unique difficulties encountered in this implementation.
3. Conducting 20 real-world trials (Tab 4) for the ablation study may lead to high variance. Do the authors have comparable results from a simulated environment?
4. Despite incorporating the consistency policy, the model remains inefficient.

**Questions:**

1. Given the large impact of different visual backbones (Sec 4.4), do the authors have comparative results indicating which foundation model (e.g., CLIP, DINOv2, SAM2) performs better?
2. How specifically do efficiency and performance metrics change after applying the consistency policy?
3. In Tab 4e, were the models re-trained when varying the FPN layers? In addition, regarding this ablation study, are there results for directly altering the resolution of high-res inputs? From current experiments, the improvements may stem from fusing two pre-trained encoders.

---

### Official Review · Reviewer_5yUm · 2024-10-31

**Soundness:** 3
**Presentation:** 3
**Contribution:** 2
**Rating:** 5
**Confidence:** 3

**Summary:**

This paper introduces a framework called "Imit-Diff" which improves the learning and efficiency of imitation learning algorithms. The method proposed emphasizes (1) using both low-resolution and high-resolution features without significantly increasing the number of tokens; (2) using semantic injection (e.g., with open-set detectors like Grounding-DINO) to guide the imitation learning with prior masks explicitly; (3) improving the sampling speed of diffusion policy with consistency models. The authors support the effectiveness of Imit-diff with real-world experiments and show advantages in cluttered scenes and task interruption.

**Strengths:**

* S1: The presentation of this paper is clear. The methods are proposed with reasonable motivation and explained clearly.

* S2: The proposed method, including dual resolution fusion, semantic injection, and consistency policy, all lead to improvement to the system, supported by the experiments.

* S3: The authors have conducted experiments on real-world robots and outperformed previous state-of-the-art ACT and diffusion policies.

**Weaknesses:**

* W1: When comparing with the previous state of the art, Imit-Diff uses a much larger vision-backbone of ConvNext + ViT-S, while ACT and Diffusion Policy only uses a ResNet-18 (L341-L342). Therefore, it is unclear whether the improvement of the method comes from a larger learnable capacity.

* W2: Table 4 might also be confusing since it shows the importance of pre-trained weights and backbones: when using ViT-S, the success rate is 30%, lower than ACT and Diffusion Policy. From my understanding, ViT-S is a more capable backbone than ResNet-18. Therefore, this table further raises concerns about the method's effectiveness and might require further clarification: Does the effectiveness come from the modules introduced by the authors or the pre-trained DINOv2 weights?

**Questions:**

The questions are mainly related to the fairness of comparison, which are in the weaknesses section. Please see above.

---

### Official Review · Reviewer_5LFU · 2024-11-02

**Soundness:** 2
**Presentation:** 2
**Contribution:** 2
**Rating:** 3
**Confidence:** 5

**Summary:**

This paper proposes the semantics guided diffusion transformer with dual resolution fusion for imitation learning. The introduced framework termed as Imit-Diff focus on semantic and fine-grained feature extraction, improving the generalization on unseen objects and environments. Imit-Diff mainly includes three key components: Dual Resolution Fusion, Semantics Injection and Consistency Policy on DiT. The proposed method outperforms some typical baselines, such as ACT and Diffusion Policy, on some real-world tasks.

**Strengths:**

* The overall presentation is relatively easy for understanding.
* The experiments show that the proposed Imit-Diff brings some improvements on some real-world tasks.

**Weaknesses:**

* The paper writing is not well prepared since there are many obvious problems. Why the proposed policy is called Rep-Diff in the Figure.1(c)? For the third paragraph of introduction, the detailed reason for introducing the proposed three modules is missed.
* The overall method indeed lacks of novelty. The ConvNext and DINOv2 are directly used as the visual encoders for the high/low-resolution inputs. Grounding DINO/MixFormerv2/MobileSAM are used for Semantics Injection. Moreover, the Consistency Policy proposed by Prasad et al (2024) is employed for few-step or single-step diffusion. The authors should clarify the contribution on the motivation. Why using ConvNext and DINOv2 for visual encoding?
* I feel the ablation study part is not inefficient. It lacks the overall ablations for the introduced three modules: Dual Resolution Fusion, Semantics Injection and Consistency Policy. I want to see the improvements of Dual Resolution Fusion (low/high resolution inputs) instead of the detailed settings, like the loss or FPN feature levels.
* The paper lacks of the comparison on the inference speed of overall framework. The comparison in Tab.5 is meaningless because CTM is only one part of Imit-Diff. I would feel the inference time is a large problem since Imit-Diff introduces so many modules such as Grounding DINO/MixFormerv2/.
* The paper only compares the Imit-Diff with ACT and Diffusion Policy. I would like to see the generalization comparision with some vision-language-action approaches such as RT-2/OpenVLA.

**Questions:**

Please see the Weaknesses section above.

---

### Official Review · Reviewer_yMh3 · 2024-11-04

**Soundness:** 3
**Presentation:** 3
**Contribution:** 2
**Rating:** 5
**Confidence:** 3

**Summary:**

This submission proposes a new diffusion-based imitation learning in robotic visuomotor control tasks.
The proposed method Imit-Diff follows the Diffusion-Transformer-based policy learning framework,
three new ideas are introduced to enhance performance.
First, Dual Resolution Fusion utilizes original-resolution and downsampled images from environment and arm-mounted cameras
 to capture global and fine-grained visual information.
Second, Semantic Injection provides input images masked by manipulation-target segments with open-vocabulary
detector and tracker.
Third, Consistency Policy by Prasad et al. (2024) is introduced for faster sampling.
Experiments are conducted using a real robot arm and the proposed method outperformed ACT and DP-T.

**Strengths:**

- The proposed method technically sounds by incorporating richer visual cues with multiple
resolution features, semantic masks, and Consistency Policy, each of which is promising to enhance diffusion-based policy learning.

- The proposed enhancement for visual input looks widely applicable and can be useful for various manipulati0on tasks in the future.

- The experiments are conducted in real hardware, unlike most of RL works in ML/AI venues.

**Weaknesses:**

- Technical novelty of the proposed method seems limited:
I think that the core of the Imit-Diff's learning mechanism is the diffusion-based policy learning,
but it is almost unchanged from Diffusion Policy's except introducing consistency-based loss.
Overall, the contribution of the work is in the system level, and might not be in the best fit to ML venue like ICLR.

- The effect of open-vocabulary vision models is not demonstrated well:
The used "unseen" manipulation targets are blocks of new colors, which seem insufficient to
assess open-set generalizability of the method.
Diverse objects are used as the clutter, but they are not similar to the targets so
it is questionable whether they make the task difficult enough.

**Questions:**

n/a

---

### Note · Authors · 2024-11-27

I have read and agree with the venue's withdrawal policy on behalf of myself and my co-authors.